# Simulation and Optimization Design of Inductive Wear Particle Sensor

**DOI:** 10.3390/s23104890

**Published:** 2023-05-19

**Authors:** Bin Fan, Lianfu Wang, Yong Liu, Peng Zhang, Song Feng

**Affiliations:** 1College of Mechanical & Electrical Engineering, Inner Mongolia Agricultural University, Hohhot 010018, China; wanglianfu1126@foxmail.com; 2National Key Laboratory of Special Vehicle Design and Manufacturing Integration Technology, Baotou 014030, China; zhangpeng9743@126.com; 3School of Advanced Manufacturing Engineering, Chongqing University of Posts and Telecommunications, Chongqing 400065, China; fengsong@cqupt.edu.cn

**Keywords:** induced electromotive force, simulation, magnetic field, structural design

## Abstract

In order to monitor the diagnosis of mechanical equipment by monitoring the metal wear particles carried in large aperture lubricating oil tubes, the simulation optimization structure design was carried out based on the traditional three-coil inductance wear particle sensor. The numerical model of electromotive force induced by the wear particle sensor was established, and the coil distance and coil turns were simulated by finite element analysis software. When permalloy is covered on the surface of the excitation coil and induction coil, the background magnetic field at the air gap increases, and the induced electromotive force amplitude generated by wear particles is increased. The effect of alloy thickness on the induced voltage and magnetic field was analyzed to determine the optimum thickness, and increase the induction voltage of the alloy chamfer detection at the air gap. The optimal parameter structure was determined to improve the detection ability of the sensor. Ultimately, by comparing the extreme values of the induced voltage of various types of sensors, the simulation determined that the minimum allowable detection of the optimal sensor was 27.5 µm ferromagnetic particles.

## 1. Introduction

During the operation of the equipment, components will wear and tear, and the metal wear particles generated and the substances, mixed with carbon slag, dust, and water during the production process, will continuously pollute the lubricating oil [1,2,3]. The pollutants in these lubricating oils accelerate the wear rate of equipment components, leading to equipment failures [4,5,6]. Additionally, wear particles are the products of friction and wear in mechanical components, and their size, shape, quantity, and other information directly reflect the wear status of equipment [7,8]. The oil detection technology analyzes and evaluates oil performance and other related indicators by detecting the state parameters of metal particles in lubricating oil and the physicochemical indicators of the oil [9,10], in order to monitor the operating status of the engine and diagnose the types and causes of faults [11,12].

Online oil monitoring methods mainly include optical, ultrasonic, and inductive methods [13,14,15]. Inductive wear particles detection technology uses a coil as an induction device, and the degree of magnetization of ferromagnetic and non-ferromagnetic particles in the magnetic field, as well as the changes in the magnetic field caused by eddy currents inside the wear particles, are reflected in the rate of change of inductance [16]. The size of metal wear particles is measured by the change in coil inductance, which is converted into a change in sensor inductance signal by a wear particle sensor [17]. The wear particle sensor distinguishes the type of wear particles based on the rate of inductance change, thereby analyzing the wear position of the equipment [18]. It has the characteristics of simple structure, high detection accuracy, and low cost, and is widely used in online oil monitoring processes [19,20,21]. In order to improve the detection sensitivity of inductive wear particle sensors, many scholars have improved the sensor structure. Ren et al. [22] used a sensor with circular excitation and two semicircular sensing coils. The same induced electromotive force of the two induction coils can be canceled, and ferrous and colored particles of 120 µm (D) and 210 µm (D) can be measured. Zhu et al. [23] proposed a 3 × 3 planar coil sensing array. By using the simultaneous sampling method, 50 µm ferromagnetic wear debris and 150 µm non-ferromagnetic wear debris can be detected using series-connected diodes to eliminate crosstalk between sensing channels. They verified that the sensor could detect debris of 50 µm in engine oil in real-time at a flow rate of 460 mL per minute. P. Muthuvel et al. [24] proposed a wear detection sensor based on a passive wireless LC sensing method. When the internal coil is close to the oil, it helps to achieve high sensitivity, and the sensor can monitor metal debris (75 µm) in thick hydraulic flow pipes (inner diameter 25.4 mm, outer diameter 80 mm, or 20 mm thick) online. Qian et al. [25] proposed a special mesh screen that can effectively eliminate interference on the three-coil induction sensor. The interference signal of water droplets with a volume of 697 mm^3^ is equivalent to the signal of iron particles with a diameter of 560 µm. This sensor can detect 353 µm (D) of black metal particles and 706 µm (D) of non-ferrous metal particles in pipelines with a diameter of 50 mm. These studies on optimizing coil structures indicate that excellent coil structures can effectively improve the induced electromotive force output signal of sensors.

The magnetic field intensity in the internal detection area of an inductive wear particle sensor can affect the magnitude of the induced electromotive force amplitude of the output signal of metal wear particles in the sensor. Many scholars have improved the internal magnetic field size of the wear particle sensor by adding specific structures. Feng et al. [26] wrapped the excitation wire coil on the iron core, which can generate a high gradient magnetic field at the air gap under constant current driving. The results showed that the sensor can detect 25 µm. The pulse width of the output signal of ferromagnetic fragments can be used to weaken or eliminate the influence of fragment velocity. Ma et al. [27] demonstrated the experimental results of dual solenoid coils with silicon steel sheets, indicating that silicon steel sheets can significantly improve the detection sensitivity of metal particles. The inductance amplitude and signal-to-noise ratio (SNR) of iron particles in the range of 60–130 µm and copper particles in the range of 120–180 µm can be increased by at least 7.0–2.4 times and 4.5–2.0 times, respectively. Zhu et al. [28] detected debris by measuring the inductance changes in two planar coils wound around a pair of ferrite cores, making the magnetic flux more dense and uniform in the sensing channel, thereby improving the sensitivity of the sensor. The static test results indicate that the debris sensor can detect 11 µm and 50 µm of iron debris in 1 mm and 7 mm diameter fluid pipelines, respectively. Wang et al. [21] used series coil mutual inductance and permalloy strong magnetic conductivity to improve the detection sensitivity of the micro channel wear particle sensor, which can detect 10–15 µm iron particles and 65–70 µm copper particles. Zheng et al. [29] designed and prepared magnetic rings using iron-based amorphous soft magnetic materials (IASM) and assembled them in large diameter 10 mm channel sensors. The magnetic ring is magnetized under the action of a high-frequency alternating magnetic field, leaving a 1 mm air gap detection area at the inner side (near the flow channel) to form an enhanced magnetic field with high gradient distribution, effectively detecting 60 µm iron particles and 160 µm copper particles. The detection error rate in the actual lubrication circuit is less than 15%. By adding magnetic materials and iron cores to enhance the magnetic field strength in the detection area, the magnetization and eddy current effects of metal particles are improved, resulting in a higher signal-to-noise ratio of the output signal and improved detection sensitivity. Therefore, it is necessary to find magnetic materials suitable for industrial production and excellent sensor structures to enhance the detection sensitivity of sensors.

In order to solve the sensitivity and stability monitoring problem of the three-coil inductive sensor under the condition of a large aperture, this paper simulates and optimizes the structure of the inductive sensor and uses finite element analysis software to simulate the magnetic field of the sensor, by adding permalloy to the coil to increase the amplitude of the induced electromotive force output by the sensor generated by wear particles, and determining the optimal structural parameters. The main contributions are as follows:Establish a mathematical model for the induced electromotive force of an inductive wear particle sensor, and use finite element analysis software to simulate the effect of coil distance on the induced voltage. Based on the simulation results, the optimal coil distance of 1.5 mm was determined;The influence of permalloy thickness and alloy air gap spacing on the sensor sensitivity was analyzed. The optimal thickness of 0.1 mm and the optimal air gap of 0.3 mm were determined through simulation. The effect of permalloy chamfer on the induced voltage was also discussed in terms of optimization;Study the induced voltage of different types of sensors wrapped with permalloy, analyze and determine the optimal sensor is the full-coverage permalloy chamfer sensor, and determine the minimum allowable detection of 27.5 µm ferromagnetic particles through simulation.

## 2. Sensor Structure, Principle Model, and Original Sensor Simulation

### 2.1. Sensor Structure and Principle

This article simulates and optimizes the structural design based on the traditional three-coil inductive wear particle sensor. The sensor structure is shown in Figure 1a. The sensor is composed of two excitation coils on both sides and an intermediate induction coil. The excitation coil and induction coil are wrapped by permalloy, and there is an air gap near the flow channel to generate a high gradient magnetic field. The alloy at the air gap is chamfered to increase the background magnetic field. The induction coil can sense the magnetic flux change caused by the passage of wear particles and output the induced electromotive force. The detection principle of the inductive wear particle sensor is shown in Figure 1b. A balanced magnetic field is formed at the center of the excitation coil by using an equal reverse current. When metal wear particles pass through the excitation coil, the magnetic field generated by magnetization or eddy current disrupts the original equilibrium magnetic field, resulting in a change in the internal magnetic flux of the coil.

### 2.2. Inductive Voltage Model

When metal particles pass through the sensor, the axial flux changes, caused by magnetization or eddy current of metal wear particles:(1)$$\Delta \varphi =\Delta B\cdot S=\pi R_{1}^{2}\left(\mu_{r}-\mu_{0}\right)V_{0}dB$$
where, $\Delta B=\left(\mu_{r}-\mu_{0}\right)V_{0}dB$ is the increment of magnetic flux caused by wear particles, r is the radius of wear particles, and $V_{0}=\frac{4}{3}\pi r^{3}$ is the volume of wear particles.

When metal particles pass through the sensor, the electromotive force output by the induction coil is as follows:(2)$$E=-N\frac{\Delta \varphi }{\Delta t}=-\pi R_{1}^{2}N\left(\mu_{r}-\mu_{0}\right)V_{0}\frac{dB}{dt}$$
wherein
(3)$$\begin{array}{cc} B=\frac{\mu_{0}NI}{2}[-(m & +x)\ln\frac{R_{2}+\sqrt{R_{2}{}^{2}+{\left(m+x\right)}^{2}}}{R_{1}+\sqrt{R_{1}{}^{2}+{\left(m+x\right)}^{2}}}+\left(W+m+x\right)\ln\frac{R_{2}+\sqrt{R_{2}{}^{2}+{\left(W+m+x\right)}^{2}}}{R_{1}+\sqrt{R_{1}{}^{2}+{\left(W+m+x\right)}^{2}}}\\ & +\left(m-x\right)\ln\frac{R_{2}+\sqrt{R_{2}{}^{2}+{\left(m-x\right)}^{2}}}{R_{1}+\sqrt{R_{1}{}^{2}+{\left(m-x\right)}^{2}}}-\left(W+m-x\right)\ln\frac{R_{2}+\sqrt{R_{2}{}^{2}+{\left(W+m-x\right)}^{2}}}{R_{1}+\sqrt{R_{1}{}^{2}+{\left(W+m-x\right)}^{2}}}] \end{array}$$

Excitation power formula:(4)$$I=I_{m}\sin\left(2\pi ft\right)$$

Equation of motion formula:(5)x=vt−W−m

When wear particles pass through the sensor, the derivative of the excitation power Formula (4) and the motion equation Formula (5) of the magnetic induction intensity B can be obtained relative to time t. The induced voltage output by the sensor can be expressed as follows:(6)$$E=-\frac{2NR_{1}^{2}\left(\mu_{r}-\mu_{0}\right)\pi^{2}r^{3}}{3}\left(\frac{dB}{dt}\right)$$
where $R_{1}$ is the inner radius of the coil, $R_{2}$ is the outer radius of the coil, and N is the number of coil turns. x is the path of fragment motion, W is the length of the coil, v is the velocity of the debris, m is the coil gap, $\;\mu_{r}$ is the wear particle permeability, and $\;\mu_{0}$ is the vacuum permeability.

### 2.3. Establishment of Original Sensor Simulation

In order to obtain an excellent model structure, finite element analysis software can be used for simulation [30]. Firstly, the coil structure of the original sensor can be optimized and simulated. The simplified simulation physical model of the original sensor is shown in Figure 2a. Using a two-dimensional axisymmetric model to model only half of the coils and metal wear particles can effectively reduce data processing and improve solution efficiency. The induction coil and excitation coil are replaced by rectangles with a distance of 1 mm between two adjacent coils. The metal wear particles adopt a semicircular structure with a particle size of 1000 µm; in terms of material selection, copper is chosen as the coil material, and iron is chosen as the wear particle material. The specific structural parameters are shown in Table 1.

In the initial interface of COMSOL, select the magnetic field frequency domain module of the AC/DC module, set the working frequency of the excitation coil to 100 kHz, and use parameterized scanning to adjust the position parameters of metal particles to simulate particle motion. As shown in Figure 2b, when there is no movement of wear particles, two equally large opposing excitation coils generate a magnetic field, and the magnetic field generated at the center of the induction coil is zero.

### 2.4. Impact of Coil Distance on Original Sensor Performance

To investigate the effect of coil distance on sensor performance, the coil parameters are modeled as shown in Table 1; the particles of iron material passing through the central axis are selected, the parameterized scanning mode is selected, and scan parameters with an interval of 0.5 mm for the distance between adjacent coils from 1 mm to 5 mm are set. Record the different coil distances and the induced voltage output by the induction coil, as shown in Figure 3.

From Figure 3, it can be seen that when the gap between the coils is 5 mm, the peak voltage output of the induction coil is the smallest when particles pass through; when the gap between the coils changes from 5 mm to 2.5 mm, the peak output voltage of the intermediate induction coil significantly increases; when the gap between the coils of the inductive sensor is 1 mm, 1.5 mm, or 2 mm, the difference in the peak output voltage changes among the three is relatively small; and from the partially enlarged image in Figure 3, it can be seen that when the gap between the coils of the induction sensor is 1.5 mm, the peak voltage output by the induction coil is the highest when particles pass through. Based on this, the gap between the coils of the inductive sensor is chosen to be 1.5 mm.

### 2.5. Effect of Coil Turns on Original Sensor Performance

When the number of turns of the coil is different, the alternating magnetic field formed by the excitation coil has different magnetization and eddy current effects on the particles, thereby affecting the induced electromotive force output by the sensor. In order to study the relationship between the number of coil turns and the induced electromotive force output by the sensor, a simulation was conducted on the relationship between the number of coil turns and the induced electromotive force. The range of coil turns was 100–600 turns, and parameter scans with an interval of 50 turns were performed. The simulation results are shown in Figure 4. As the number of turns increases, the induced electromotive force output by the sensor gradually increases.

## 3. Covered with Permalloy Type Wear Particle Sensor

### 3.1. Effect of Exciter Coil Covered with Permalloy on Sensor Performance

In order to study the influence of the permalloy-covered excitation coils on the sensor detection sensitivity, the simulation analysis was carried out with the permalloy-covered excitation coils on both sides. As shown in Figure 5, when the excitation coil covers the permalloy with a relative permeability of 10,000, an air gap of 0.2 mm is left on the permalloy near the inner side of the flow channel. The magnetic lines of force generated by the excitation coil gather through the 0.2 mm air gap and enter the flow channel area of the sensor, which will form a high gradient magnetic field in the permalloy air gap area. Compared to the extreme value of 0.0259 T of the magnetic flux density of the original sensor in Figure 2b, the extreme value of the magnetic flux density of permalloy covered by the excitation coil is increased to 5.37 T, which proves that adding permalloy with high permeability to the sensor can improve its background magnetic induction strength.

Under a strong magnetic field, the magnetization of ferromagnetic wear particles further increases, and the eddy current of non-ferrous wear particles also increases, leading to obvious changes in coil inductance. As shown in Figure 6, the maximum induced voltage of the original sensor is 0.23479 V, and the maximum induced voltage of the permalloy sensor covered by the excitation coil is 0.52363 V, showing a significant enhancement effect.

### 3.2. The Influence of Air Gap Spacing on Sensor Performance

In order to study the effect of air gap spacing on sensor sensitivity, the permalloy air gap spacing near the inner side of the flow channel was analyzed. The research content selected a parameterized scanning mode, and conducted parameter scans with an interval of 0.1 mm for air gap intervals ranging from 0.2 mm to 0.6 mm. The induced voltage output by the induction coil with different air gap intervals of the alloy was recorded, as shown in Figure 7.

From Figure 7, it can be seen that the difference in induced voltage peak value change caused by different air gap intervals of permalloy inside the flow channel is small. It can be seen from the locally enlarged drawing that when the air gap spacing of permalloy is 0.6 mm, the peak voltage output by the induction coil is the smallest when particles pass through; When the air gap spacing of permalloy is 0.3 mm, the peak voltage output of the induction coil is the largest when particles pass through.

### 3.3. Effect of Full-Coverage Permalloy on Sensor Performance

In order to study the influence on the magnetic field and sensitivity of the sensor after the excitation coil and induction coil are fully covered with permalloy, the analysis is carried out when the excitation coil and induction coil are covered with permalloy. According to the optimal air gap length obtained above, as shown in Figure 8, the upper, lower, and outer sides of the induction coil are covered with permalloy with a relative permeability of 10,000, and the excitation coil has an air gap of 0.3 mm near the flow channel inside the permalloy. As the length of the air gap increases, the extreme value of the magnetic flux density of the exciter coil covered with permalloy in Figure 5 is 5.37 T, and the extreme value of the magnetic flux density of the full-coverage permalloy sensor in Figure 8 drops to 4.42 T.

As shown in Figure 9a, the magnetic field of the flow channel of the full-coverage permalloy sensor is separately simulated, and the magnetic field at the flow channel corresponding to the air gap is stepped. The magnetic flux density diagram of the corresponding channel radial section A-B at the air gap is shown in Figure 9b. The magnetic flux density of the corresponding channel at the air gap is above 0.02 T compared with the extreme value of 0.0259 T magnetic flux density of the original sensor in Figure 2b. The magnetic flux density of the channel of the full-coverage permalloy sensor is enhanced.

In the full-coverage permalloy sensor, the permalloy covering on the upper, lower, and outer sides of the induction coil leads to obvious changes in the voltage of the induction coil. As shown in Figure 10, the maximum induced voltage of the original sensor is 0.23479 V, the maximum induced voltage of the full-coverage permalloy sensor is 0.63134 V, and the induced voltage of the excitation coil covered with permalloy sensor is increased by 20.5%.

### 3.4. The Effect of Alloy Thickness on Sensor Performance

In order to study the influence of permalloy thickness on sensor sensitivity, the excitation coil coverage permalloy with different thicknesses was analyzed. The research content selects the parametric scanning mode, carries out the parameter scanning with an interval of 0.1 mm for the thickness of permalloy from 0.1 mm to 0.5 mm, and records the induction voltage of different alloy thicknesses and the output of the induction coil. The results are shown in Figure 11.

From Figure 11, it can be seen that the difference between the peak changes in induced voltage produced by permalloy with a package thickness of 0.1 mm–0.5 mm is small. It can be seen from the locally enlarged drawing that when the alloy thickness of permalloy is 0.5 mm, the peak voltage output by the induction coil is the smallest when particles pass through; when the alloy thickness of permalloy is 0.1 mm, the peak voltage output of the induction coil is the largest when particles pass through.

### 3.5. Effect of Full-Coverage Permalloy Chamfer on Sensor Performance

In order to study the influence of permalloy chamfer on the sensor performance at the air gap after the sensor is covered with permalloy, the chamfer at the air gap was studied when the excitation coil was covered with permalloy. The model structure was set to the optimized structure obtained earlier. As shown in Figure 12, the excitation coil is chamfered at the permalloy air gap interval near the inner side of the flow channel. Figure 7 shows that the magnetic flux density extreme value of the full-coverage permalloy sensor is 4.42 T, and the magnetic flux density extreme value of the full-coverage permalloy chamfer is reduced to 4.2 T.

Magnetic field simulation was carried out for the channel of the full-coverage permalloy chamfer sensor; as shown in Figure 13a, the corresponding channel magnetic flux density extreme value at the air gap was 0.329 T. This is compared with Figure 9a, where the magnetic flux density extreme value of the full-coverage permalloy sensor was 0.333 T, and the magnetic flux density reduced by 1.2%. The magnetic flux density pattern of the corresponding channel radial section A-B at the air gap is shown in Figure 13b, and the enhancement effect of the magnetic flux density pattern of the full-coverage permalloy sensor is not obvious compared with Figure 9b.

In the full-coverage permalloy chamfer sensor, the permalloy chamfer at the air gap causes a weak change in the induction coil voltage. As shown in Figure 14, the maximum induced voltage of the full-coverage permalloy chamfer sensor is 0.63154 V, and the maximum induced voltage of the full-coverage permalloy sensor is 0.63134 V, which is 0.036% higher than that of the full-coverage permalloy sensor. By comparing the maximum and minimum coil voltage in Table 2, it can be seen that the coil voltage generated by the full-coverage permalloy chamfer sensor is the maximum.

### 3.6. Minimum Allowable Detection

In order to study the detection accuracy of the full-coverage permalloy chamfer, a parameter scanning simulation was carried out for particle size. The particle size ranges from 50 µm to 500 µm, and the parameters with an interval of 50 µm are scanned to record the induced voltage output by the induction coil of each particle size. The simulation results are shown in Figure 15a. With the increase in particle size, the voltage peak value of the induction coil output increases gradually.

The parameters of particle size from 37.5 µm to 47.5 µm were scanned with an interval of 2.5 µm to record the induced voltage output by the intermediate coil of each particle size. The simulation results are shown in Figure 15b. As the particle size decreases, obvious noise signals begin to appear in the induction coil. It can be seen from Figure 16a that the detection accuracy is low, and there is a large amount of noise when the particle size of 25 µm is detected after permalloy is wrapped in chamfer. The induced voltage signal is obvious from the particle size of 27.5 µm in Figure 16b. It can be seen from Figure 15c–f that the induced voltage signal increases gradually with the increase in the particle size of the tiny wear particles. Subsequent research can use circuits to remove signal noise and extract weak signals from the signals of wear particles.

## 4. Discussion

In order to improve the detection accuracy of inductive wear particle sensors, many scholars use different methods to overcome the influence of different factors on the sensor in practical work. Sensors are susceptible to weak signals and signal noise during signal transmission, resulting in incorrect differentiation between wear size and type. In this study, there is a noise signal in the detection signal of small particles. The possible reason for this situation is that the disturbance effect of permalloy on the magnetic field is enhanced, and the other possible reason is that the parameter setting in the simulation process is not ideal. Researchers have improved detection circuits and data algorithms to suppress signal noise and enhance wear output signals after signal conversion. Zhu et al. [23] applied parallel L-C-R resonance to the sensing coil of a high-throughput wear debris sensor composed of 3 × 3 sensing channels, eliminating crosstalk between each sensing channel by connecting a diode in series with each channel, reducing the influence of coil resonance on induced voltage by adding external circuits. Based on the problem of signal aliasing generated by inductive sensor fragment detection, Li et al. [31] studied the separation of superimposed wear particles in radial magnetic field (RMF) wear sensors using degradation decomposition estimation technology. The method of combining bandpass filter and DUET is used to separate the signal source from the mixed signal, and the morphology and quantity of wear particles are accurately separated and calculated. Due to the tendency of small metal particles under aggregation to misjudge the type of fault in metal wear sensors, understanding the signal aliasing caused by the aggregation effect of multiple wear particles is beneficial for improving the structure of the sensor.

The author team of this study optimized the sensor through finite element analysis software. Simulations to avoid material loss during the actual preparation process were carried out, providing a useful method for developing high-quality sensors with excellent performance. Compared to the minimum detectable ferromagnetic particle (sphere radius), the sensor designed in this article has a minimum detectable particle (sphere radius) of 27.5 µm, which is better than the MS3505 online oil pollution monitor produced by the Gastops Company’s minimum detectable particle (sphere radius) of 62.5 µm. In this study, the author explores the effects of structural parameters such as coil distance, alloy thickness, air gap spacing, and chamfer structure on induced voltage. In subsequent research work, the authors will study the influence of coil resonance circuits and mixed factors of wear particles on the performance of the sensor, and verify through experiments the optimal structural parameters obtained for the sensor.

## 5. Conclusions

This article proposes improving the detection sensitivity by improving the coil structure and adding magnetic materials using finite element analysis software to determine the optimal structural parameters and alloy addition method. It proposes a mathematical model for sensor-induced electromotive force, and simulates the influence of structural factors such as coil distance on induced voltage to determine the optimal coil distance of 1.5 mm. Then, the influence of alloy thickness and alloy air gap spacing on sensor sensitivity was studied by adding magnetic material permalloy. The optimal thickness of 0.1 mm and the optimal air gaps of 0.3 mm were determined through simulation, and the alloy at the air gap was chamfered to slightly improve the induced voltage. Finally, by comparing the extreme values of the induced voltage of various types of sensors, the simulation determined that the minimum allowable detection of the optimal sensor was 27.5 µm ferromagnetic particles.

## Figures and Tables

**Figure 1 sensors-23-04890-f001:**
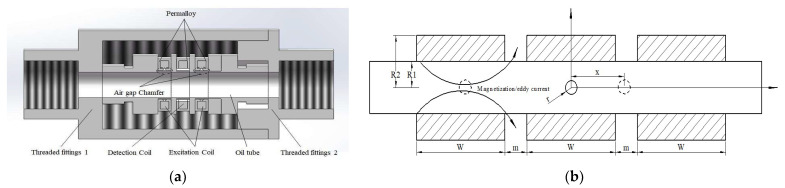
(**a**) Structure diagram of wear particle sensor; (**b**) schematic diagram of the original wear particle sensor detection.

**Figure 2 sensors-23-04890-f002:**
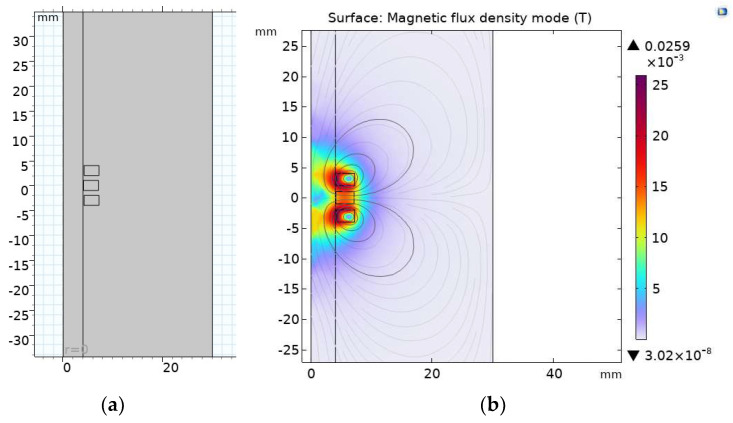
(**a**) 2D axisymmetric model; (**b**) the zero magnetic field at the center.

**Figure 3 sensors-23-04890-f003:**
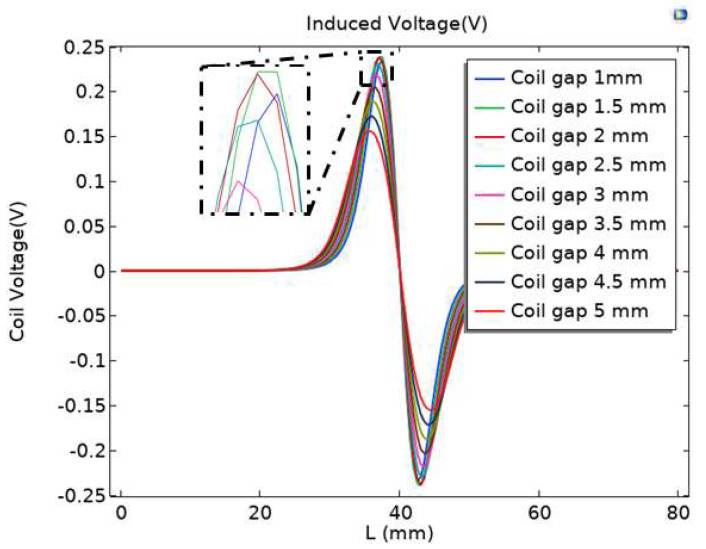
Induced electromotive force at different coil distances.

**Figure 4 sensors-23-04890-f004:**
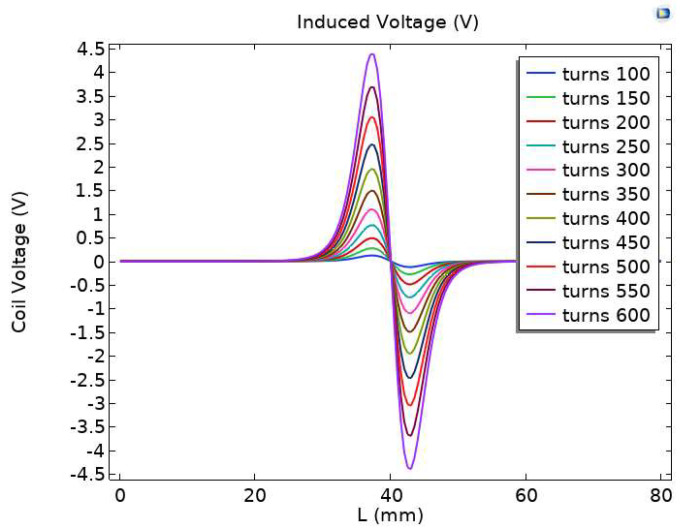
Induced electromotive force under different turns.

**Figure 5 sensors-23-04890-f005:**
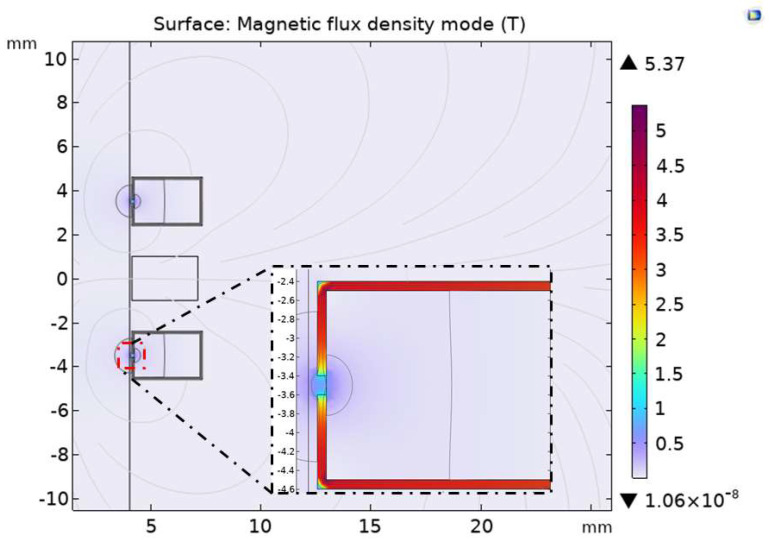
Magnetic flux density of exciter coil covered with permalloy.

**Figure 6 sensors-23-04890-f006:**
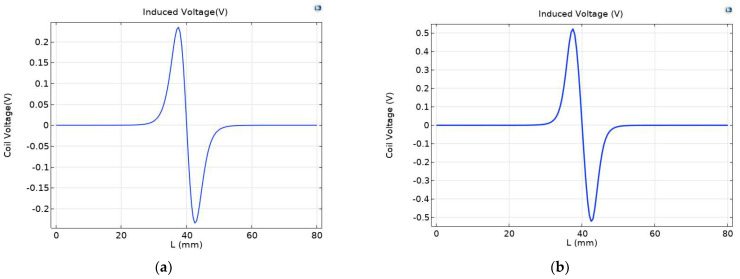
Comparison of induced voltage between (**a**) the original sensor and (**b**) the exciter coil covered with permalloy on sensors.

**Figure 7 sensors-23-04890-f007:**
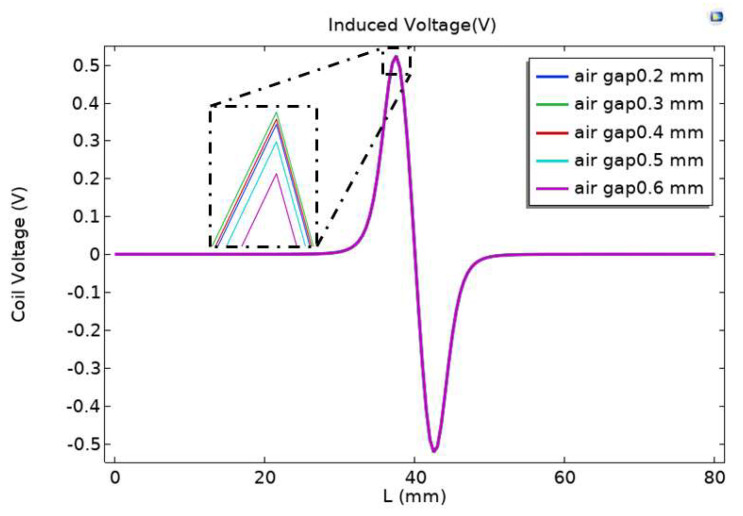
Impact of air gap on sensor performance.

**Figure 8 sensors-23-04890-f008:**
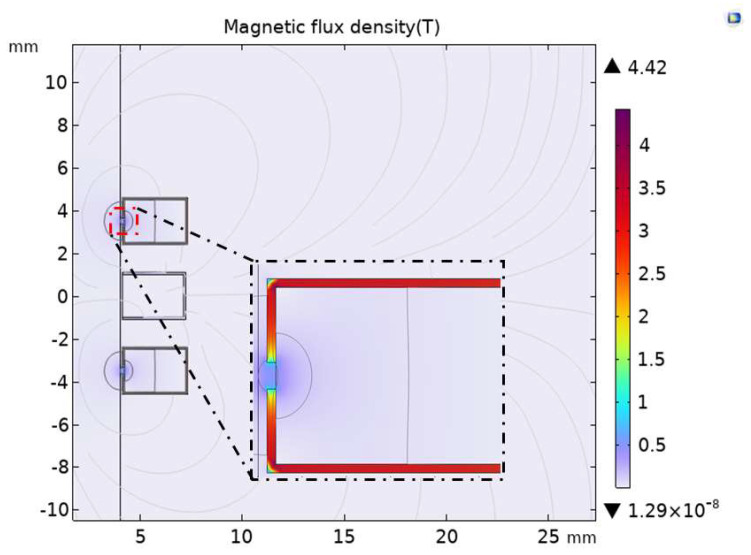
Magnetic flux density of full-coverage permalloy sensor.

**Figure 9 sensors-23-04890-f009:**
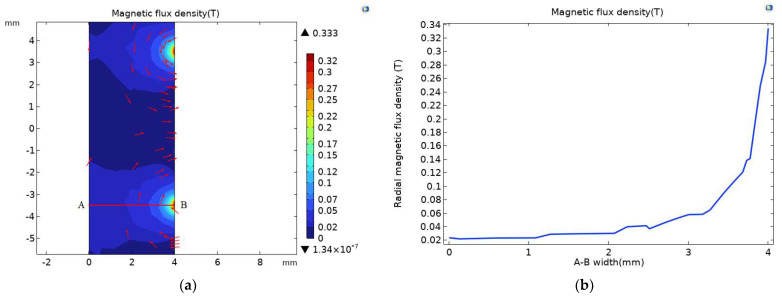
(**a**) Magnetic flux density of full-coverage permalloy sensor channel; (**b**) magnetic flux density pattern at channel A-B line.

**Figure 10 sensors-23-04890-f010:**
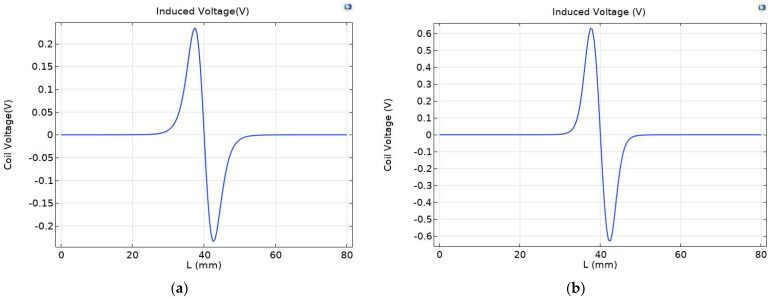
Comparison of induced voltage between (**a**) the original sensor and (**b**) the full-coverage sensor.

**Figure 11 sensors-23-04890-f011:**
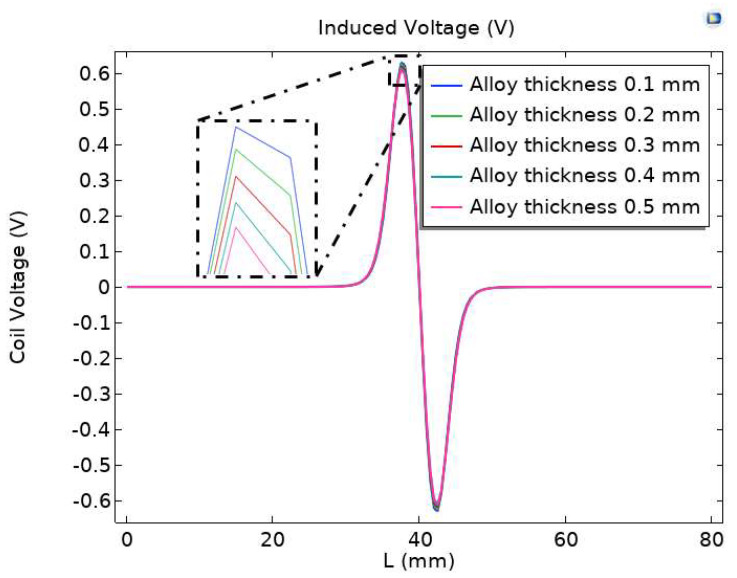
Effect of alloy thickness on sensor performance.

**Figure 12 sensors-23-04890-f012:**
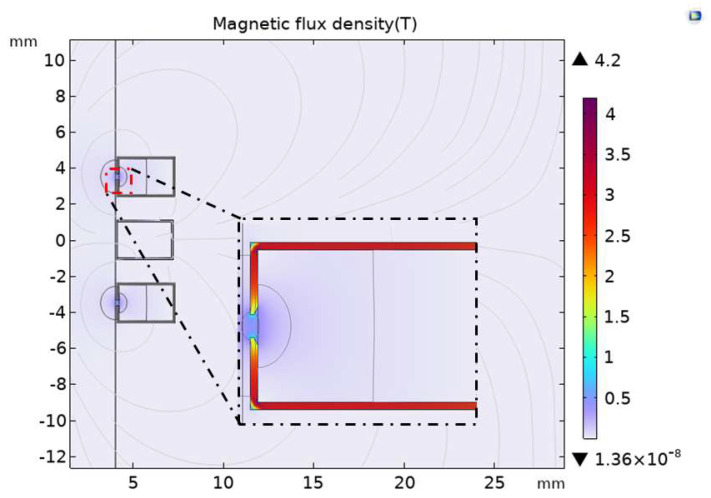
Magnetic flux density of chamfer full-coverage permalloy sensor.

**Figure 13 sensors-23-04890-f013:**
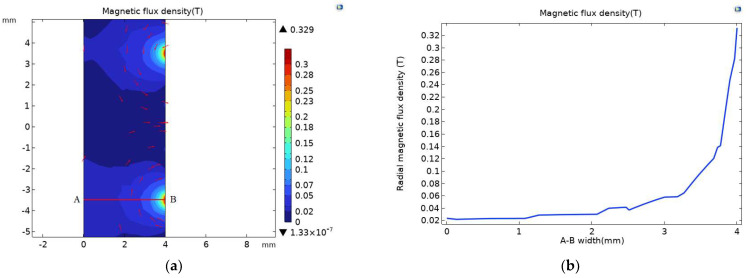
(**a**) The magnetic flux density of the chamfer fully covering the permalloy sensor channel; (**b**) magnetic flux density pattern at channel A-B line.

**Figure 14 sensors-23-04890-f014:**
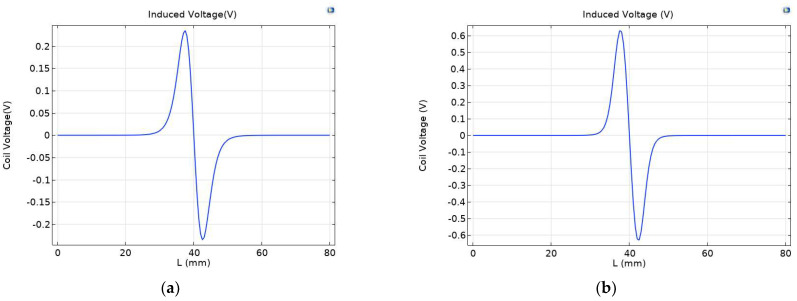
Comparison of induced voltage between (**a**) the original sensor and (**b**) the chamfer full-coverage sensor.

**Figure 15 sensors-23-04890-f015:**
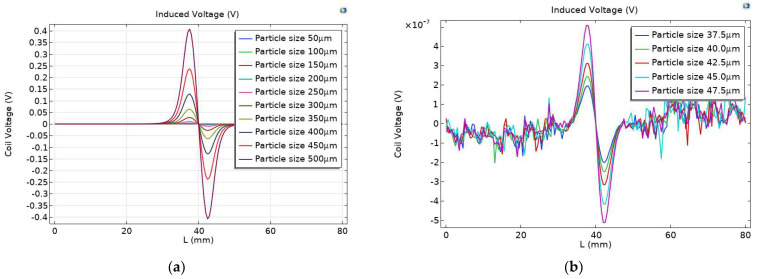
The induced voltage of different particle sizes for (**a**) 50–500 µm particle size and (**b**) 37.5–47.5 µm particle size.

**Figure 16 sensors-23-04890-f016:**
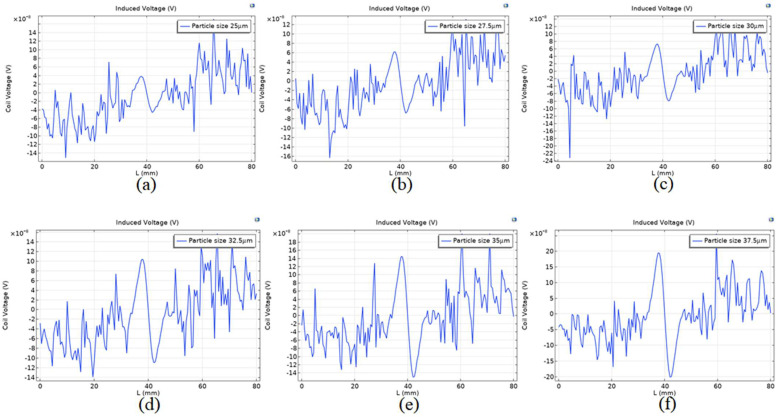
The induced voltage of different particle sizes at (**a**) 25 µm, (**b**) 27.5 µm, (**c**) 30 µm, (**d**) 32.5 µm, (**e**) 35 µm, and (**f**) 37.5 µm.

**Table 1 sensors-23-04890-t001:** Sensor structural parameters.

Excitation Current	Tube Radius	Wire Diameter	Number of Turns of Induction/Excitation Coils	Excitation Frequency	Coil Gap
1 A	4 mm	0.2 mm	140	100 kHz	1 mm

**Table 2 sensors-23-04890-t002:** Comparison of maximum and minimum coil voltages for various types of sensors.

Sensor Type	Maximum Value: Coil Voltage	Minimum Value: Coil Voltage
Full-coverage permalloy chamfer	0.63154 V	−0.63098 V
Full-coverage permalloy	0.63134 V	−0.63083 V
Exciting coil covered with permalloy chamfer	0.52396 V	−0.52226 V
Exciter coil covered with permalloy	0.52363 V	−0.52198 V
Original sensor	0.23479 V	−0.23401 V

## Data Availability

Not applicable.

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
