# Peer review of "Simulation and Optimization Design of Inductive Wear Particle Sensor"

_sensors, 2023, doi:10.3390/s23104890_

Round 1

Reviewer 1 Report

This paper presents the optimization of a sensor structure. The magnetic field is simulated to improve the detection capability of the sensor. The optimal coating thickness of the alloy is determined by studying the relationship between alloy thickness and induced voltage and magnetic field. The paper is well written. Some detailed comments are provided below.

(1)    In equation (1), on the left side, S is area, while on the right side, pi*R*R is area and V0 is volume. Please check.

(2)    Fig.1 is not clear enough, the authors could provide more figures from different views, so that the readers could understand the structure more easily.

(3)    In Fig.3, can the authors explain why the voltage has two poles around the point of L=40mm?

(4)    Fig.16 is not clear. Please improve.

(5)    The authors only provide simulation results, it will better if measurement results are given.

Reviewer 2 Report

The work is devoted to improving the efficiency of the wear sensor. The work has a logically correct structure; the mathematical description does not cause any comments. The text is hard to read. I am not a native speaker and cannot judge, but many parts of the article are reminiscent of reading an instruction manual and not a scientific article.

Notes:

1. Line 99-106. The author needs to reformulate the paragraph and indicate the exact statement of the problem. What is written from 25 to 98 is understandable ... but what is the result? Why do you need research.

2. Line 131-137. You are positioning the article as the development of a new device to improve efficiency. In this section, you need to show what you offer fundamentally new. The sensor you are considering is traditional. What's your take on this sensor? The author needs to add text indicating structural changes.

3. Line 141-156. It feels like writing out an instruction, not a scientific article. This text needs to be revised.

4. Figure 13, 14. I agree. But resonance between the coils is possible. It is necessary to supplement the text with the question of resonance.

5. The article does not have a "Discussion" section. You need to add a section before the conclusion. In this section, it is necessary to talk about the positive and negative aspects of the study. Show ways to apply this study. The author is recommended to read and include in the review of this section the article https://doi.org/10.3390/en15239001 in which the authors directly indicate the need for such a sensor to analyze the technical condition of the cell. Also, the author is recommended to read and include in the review of this section the studies doi: 10.1109/ElConRus54750.2022.9755852 and doi: 10.1109/ElConRus51938.2021.9396250 in them, the authors show the need to use this sensor for diagnosing oil and gas equipment.

6. Line 340-387. According to the requirements for scientific publications, the "References" section should contain from 30-40 sources. The author needs to expand the list of sources at the expense of the above sources and those found in search engines.

Conclusion. The article can be recommended for publication after the elimination of these shortcomings.

Round 2

Reviewer 1 Report

Thanks to the authors for the revision, my concerns have been addressed.

Reviewer 2 Report

All comments have been removed. I recommend the work for publication.